# Fast and Accurate Sequence Labeling with Iterated Dilated Convolutions

## Abstract

Bi-directional LSTMs have emerged as a standard method for obtaining per-token vector representations serving as input to various token labeling tasks (whether followed by Viterbi prediction or independent classification). This paper proposes an alternative to Bi-LSTMs for this purpose: iterated dilated convolutional neural networks (ID-CNNs), which have better capacity than traditional CNNs for large context and structured prediction. We describe a distinct combination of network structure, parameter sharing and training procedures that is not only more accurate than Bi-LSTM-CRFs, but also 8x faster at test time on long sequences. Moreover, ID-CNNs with independent classification enable a dramatic 14x test-time speedup, while still attaining accuracy comparable to the Bi-LSTM-CRF. We further demonstrate the ability of ID-CNNs to combine evidence over long sequences by demonstrating their improved accuracy on whole-document (rather than per-sentence) inference. Unlike LSTMs whose sequential processing on sentences of length $N$ requires $O(N)$ time even in the face of parallelism, IDCNNs permit fixed-depth convolutions to run in parallel across entire documents. Today when many companies run basic NLP on the entire web and large-volume traffic, faster methods are paramount to saving time and energy costs.

## 1 Introduction

In order to democratize large-scale NLP and information extraction, we require fast, resource-efficient methods for sequence tagging tasks such as part-of-speech tagging and named entity recognition (NER). Speed is not sufficient of course: they must also be expressive enough to tolerate the tremendous lexical variation in input data.

The massively parallel computation facilitated by GPU hardware has led to a surge of successful neural network architectures for sequence labeling (Ling et al., 2015; Ma and Hovy, 2016; Chiu and Nichols, 2016; Lample et al., 2016). While these models are expressive and accurate, they fail to fully exploit the parallelism opportunities of a GPU, and thus their speed is limited. Specifically, they employ either recurrent neural networks (RNNs) for feature extraction, or Viterbi inference in a structured output model, both of which require sequential computation across the length of the input.

Instead, parallelized runtime independent of the length of the sequence saves time and energy costs, maximizing GPU resource usage and minimizing the amount of time it takes to train and evaluate models. Convolutional neural networks (CNNs) provide exactly this property (Kim, 2014; Kalchbrenner et al., 2014). Rather than composing representations incrementally over each token in a sequence, they apply filters in parallel across the entire sequence at once. Their computational cost grows with the number of layers, but not the input size, up to the memory and threading limitations of the hardware. This provides, for example, audio generation models that can be trained in parallel (van den Oord et al., 2016).

Despite the clear computational advantages of CNNs, RNNs have become the standard method for composing deep representations of text. This is because a token encoded by a bidirectional RNN will incorporate evidence from the entire input sequence, but the CNN's representation is limited by the *receptive field* of the architecture. Specifi-

cally, in a network composed of a series of stacked convolutional layers of convolution width $w$, the number $r$ of context tokens incorporated into a token's representation at a given layer $l$, is given by $r = 2l(w-1)+1$. The number of layers required to incorporate the entire input context grows linearly with the length of the sequence. To avoid this scaling, one could pool representations across the sequence, but this is not appropriate for sequence labeling, since it reduces the output resolution of the representation.

In response, this paper presents an application of *dilated convolutions* (Yu and Koltun, 2016) for sequence labeling (Figure 1). For dilated convolutions, the receptive field can grow exponentially with the depth, with no loss in resolution at each layer and with a modest number of parameters to estimate. Like typical CNN layers, dilated convolutions operate on a sliding window of context over the sequence, but unlike conventional convolutions, the context need not be consecutive; the dilated window skips over every dilation width $d$ inputs. By stacking layers of dilated convolutions of exponentially increasing dilation width, we can expand the size of the receptive field to cover the entire length of most sequences using only a few layers: The size of the receptive field for a token at layer $l$ is now given by $2^{l+1}-1$. More concretely, just four stacked dilated convolutions of width 3 produces token representations with a receptive field of 31 tokens – longer than the average sentence length (23) in the Penn TreeBank.

Our overall *iterated dilated CNN* architecture (ID-CNN) repeatedly applies the same block of dilated convolutions to token-wise representations. This parameter sharing prevents overfitting and also provides opportunities to inject supervision on intermediate activations of the network. Similar to models that use RNN features, the ID-CNN provides two methods for performing prediction: we can predict each token's label independently, or by running Viterbi inference in a chain structured graphical model.

In experiments on CoNLL 2003 and Ontonotes 5.0 English NER, we demonstrate significant speed gains of our ID-CNNs over various recurrent models, while maintaining similar F1 performance. When performing prediction using independent classification, the ID-CNN consistently outperforms a bidirectional LSTM (Bi-LSTM), and performs on par with inference in a CRF

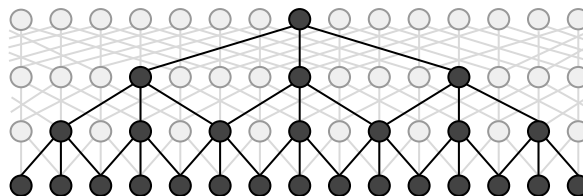

Figure 1: A dilated CNN block with maximum dilation width 4 and filter width 3. Neurons contributing to a single highlighted neuron in the last layer are also highlighted.

with features extracted by a Bi-LSTM (Bi-LSTM-CRF). As a feature extractor for a CRF, our model out-performs the Bi-LSTM-CRF. We also apply ID-CNNs to entire documents, where independent token classification is more accurate than the Bi-LSTM-CRF while decoding almost $8\times$ faster. The clear accuracy gains resulting from incorporating broader context suggest that these models could similarly benefit many other context-sensitive NLP tasks which have until now been limited by the computational complexity of existing context-rich models.[1]

## 2 Background

### 2.1 Conditional Probability Models for Tagging

Let $x = [x_1, \ldots, x_T]$ be our input text and $y = [y_1, \ldots, y_T]$ be per-token output tags. Let $D$ be the domain size of each $y_i$. We predict the most likely $y$, given a conditional model $P(y|x)$.

This paper considers two factorizations of the conditional distribution. First, we have

$$P(y|x) = \prod_{t=1}^{T} P(y_t|F(x)), \qquad (1)$$

where the tags are conditionally independent given some features for x. Given these features, $O(D)$ prediction is simple and parallelizable across the length of the sequence. However, feature extraction may not necessarily be parallelizable. For example, RNN-based features require iterative passes along the length of $x$.

We also consider a linear-chain CRF model that couples all of $y$ together:

$$P(y|x) = \frac{1}{Z_x} \prod_{t=1}^{T} \psi_t(y_t|F(x))\psi_p(y_t, y_{t-1}), \quad (2)$$

---

[1]Our implementation in TensorFlow (Abadi et al., 2015) is available at: https://github.com/anonymized

where $\psi_t$ is a local factor, $\psi_p$ is a pairwise factor that scores consecutive tags, and $Z_x$ is the partition function (Lafferty et al., 2001). To avoid overfitting, $\psi_p$ does not depend on the timestep $t$ or the input $x$ in our experiments. Prediction in this model requires global search using the $O(D^2T)$ Viterbi algorithm.

CRF prediction explicitly reasons about interactions among neighboring output tags, whereas prediction in the first model compiles this reasoning into the feature extraction step (Liang et al., 2008). The suitability of such compilation depends on the properties and quantity of the data. While CRF prediction requires non-trivial search in output space, it can guarantee that certain output constraints, such as for IOB tagging (Ramshaw and Marcus, 1999), will always be satisfied. It may also have better sample complexity, as it imposes more prior knowledge about the structure of the interactions among the tags (London et al., 2016). However, it has worse computational complexity than independent prediction.

## 3   Dilated Convolutions

CNNs in NLP are typically one-dimensional, applied to a sequence of vectors representing tokens rather than to a two-dimensional grid of vectors representing pixels. In this setting, a convolutional neural network layer is equivalent to applying an affine transformation, $W_c$ to a sliding window of width $r$ tokens on either side of each token in the sequence. Here, and throughout the paper, we do not explicitly write the bias terms in affine transformations. The sliding-window representation $c_t$ for each token $x_t$ is:

$$c_t = \bigoplus_{k=0}^{r} x_{t \pm k}, \qquad (3)$$

where $\oplus$ is vector concatenation.

Dilated convolutions perform the same operation, except rather than transforming adjacent inputs, the convolution is defined over a wider receptive field by skipping over $\delta$ inputs at a time, where $\delta$ is the dilation width. We define the dilated convolution operator:

$$c_t = \bigoplus_{k=0}^{r} x_{t \pm k\delta}. \qquad (4)$$

A dilated convolution of width 1 is equivalent to a simple convolution. Using the same number of

parameters as a simple convolution with the same radius, the $\delta > 1$ dilated convolution incorporates broader context into the representation of a token than a simple convolution.

Finally, Lei et al. (2015) present a CNN variant where convolutions adaptively skip neighboring words. While the flexibility of this model is powerful, its adaptive behavior is not well-suited to GPU acceleration.

### 3.1   Multi-Scale Context Aggregation

We can leverage the ability of dilated convolutions to incorporate global context without losing important local information by stacking dilated convolutions of increasing width. First described for pixel classification in computer vision, Yu and Koltun (2016) achieve state-of-the-art results on image segmentation benchmarks by stacking dilated convolutions with exponentially increasing rates of dilation, a technique they refer to as *multi-scale context aggregation*. By feeding the outputs of each dilated convolution as the input to the next, increasingly non-local information is incorporated into each pixel's representation. Performing a dilation-1 convolution in the first layer ensures that no pixels within the receptive field of any pixel are excluded. By doubling the dilation width at each layer, the size of the receptive field grows exponentially while the number of parameters grows only linearly with the number of layers, so a pixel representation quickly incorporates rich global evidence from the entire image.

## 4   Iterated Dilated CNNs

Stacked dilated CNNs can easily incorporate global information from a whole sentence or document. For example, with a radius of 1 and 4 layers of dilated convolutions, the receptive field of each token is width 31, which exceeds the average sentence length (23) in the Penn TreeBank corpus. With a radius of size 2 and 8 layers of dilated convolutions, the receptive field exceeds 1,000 tokens, long enough to encode an many full documents.

Unfortunately, simply increasing the depth of stacked dilated CNNs causes considerable overfitting in our experiments. In response, we present Iterated Dilated CNNs (ID-CNNs), which instead iterate a small series of dilated convolutions. Repeatedly employing the same parameters in a recurrent fashion provides both broad receptive fields and desirable generalization capabil-

ities. We also obtain significant accuracy gains with a training objective that strives for accurate labeling after each iterate, allowing follow-on iterates to observe and resolve dependency violations.

## 4.1 Model Architecture

ID-CNNs contain repeated blocks of several convolutional layers. The network takes as input a sequence of $T$ vectors $\mathbf{x_t}$ of dimension $d_w$, and outputs a sequence of per-class scores $\mathbf{h_t}$, which serve either as the local conditional distributions of the model (1) or the local factors $\psi_t$ of model (2).

The first layer in the network is a dilation-1 convolution $D_1^{(1)}$ that transforms the input to a representation $\mathbf{i_t}$ of dimension $d_c$:

$$\mathbf{i_t} = D_1^{(0)}\mathbf{x_t} \tag{5}$$

Next, $L_c-1$ layers of dilated convolutions of exponentially increasing dilation width are applied to $\mathbf{i_t}$, folding in increasingly broader context into the embedded representation of $\mathbf{x_t}$ at each layer, followed by a dilation-1 convolution. Let $r()$ denote a ReLU activation function (Glorot et al., 2011). Beginning with $\mathbf{c_t}^{(0)} = \mathbf{i_t}$ we define the stack of layers with the following recurrence:

$$\mathbf{c_t}^{(L_c-1)} = r\left(D_{2^{L_c-2}}^{(1)}\mathbf{c_t}^{(L_c-2)}\right)$$
$$\mathbf{c_t}^{(L_c)} = r\left(D_1^{(3)}\mathbf{c_t}^{(L_c-1)}\right) \tag{6}$$

We refer to this stack of dilated convolutions as a *block* $B$, which has output resolution equal to the input resolution. To incorporate even broader context without over-fitting, we avoid making $B$ deeper, and instead iteratively apply $B$ $L_b$ times, which introduces no extra parameters. Starting with $\mathbf{b_t}^{(1)} = B\left(\mathbf{i_t}\right)$:

$$\mathbf{b_t}^{(L_b)} = B\left(\mathbf{b_t}^{(L_b-1)}\right) \tag{7}$$

We apply a simple affine transformation $W_c$ to this final representation to obtain per-class scores for each token $\mathbf{x_t}$:

$$\mathbf{h_t} = W_c\mathbf{b_t}^{(L_b)} \tag{8}$$

## 4.2 Training

Our main focus is to apply the ID-CNN as feature extraction for the first conditional model described in Sec. 2.1, where tags are conditionally independent given deep features, since this will enable prediction that is parallelizable across the length of the input sequence. Here, maximum likelihood training is straightforward because the likelihood decouples into the sum of the likelihoods of independent logistic regression problems for every tag, with natural parameters given by (8):

$$\frac{1}{T}\sum_{t=1}^{T}\log P(y_t \mid \mathbf{h_t}) \tag{9}$$

We can also use the ID-CNN as input features for the CRF model (2), where the partition function and its gradient are computed using the forward-backward algorithm.

We next present an alternative training method that helps bridge the gap between these two techniques. Sec. 2.1 identifies that the CRF has preferable sample complexity and accuracy since prediction directly reasons in the space of structured outputs. In response, we compile some of this reasoning in output space into ID-CNN feature extraction. Instead of explicit reasoning over output labels during inference, we train the network such that each block is predictive of output labels. Subsequent blocks learn to correct dependency violations of their predecessors, refining the final sequence prediction.

To do so, we first define predictions of the model after each of the $L_b$ applications of the block. Let $\mathbf{h_t^b}$ be the result of applying the matrix $W_c$ from (8) to the output of the block $b$, or the initial word embeddings in the case of $b = 0$. We minimize the average of the losses for each application of the block:

$$\frac{1}{B}\sum_{b=0}^{B}\frac{1}{T}\sum_{t=1}^{T}\log P(y_t \mid \mathbf{h_t^b}). \tag{10}$$

By rewarding accurate predictions after each application of the block, we learn a model where later blocks are used to refine initial predictions. The loss also helps reduce the vanishing gradient problem (Hochreiter, 1998) for deep architectures. Such an approach has been applied in a variety of contexts for training very deep networks in computer vision (Romero et al., 2014; Szegedy et al., 2015; Lee et al., 2015; Gülçehre and Bengio, 2016), but not to our knowledge in NLP.

We apply dropout (Srivastava et al., 2014) to the raw inputs $\mathbf{x_t}$ and to each block's output $\mathbf{b_t}^{(b)}$ to help prevent overfitting. The version of dropout typically used in practice has the undesirable property that the randomized predictor used at train

time differs from the fixed one used at test time. Ma et al. (2017) present *dropout with expectation-linear regularization*, which explicitly regularizes these two predictors to behave similarly. All of our best reported results include such regularization. This is the first investigation of the technique's effectiveness for NLP, including for RNNs. We encourage its further application.

## 5   Related work

The state-of-the art models for sequence labeling include an inference step that searches the space of possible output sequences of a chain-structured graphical model, or approximates this search with a beam (Collobert et al., 2011; Weiss et al., 2015; Lample et al., 2016; Ma and Hovy, 2016; Chiu and Nichols, 2016). These outperform similar systems that use the same features, but independent local predictions. On the other hand, the greedy *sequential prediction* (Daumé III et al., 2009) approach of Ratinov and Roth (2009), which employs lexicalized features, gazetteers, and word clusters, outperforms CRFs with similar features.

LSTMs (Hochreiter and Schmidhuber, 1997) were used for NER as early as the CoNLL shared task in 2003 (Hammerton, 2003; Tjong Kim Sang and De Meulder, 2003). More recently, a wide variety of neural network architectures for NER have been proposed. Collobert et al. (2011) employ a one-layer CNN with pre-trained word embeddings, capitalization and lexicon features, and CRF-based prediction. Huang et al. (2015) achieved state-of-the-art accuracy on part-of-speech, chunking and NER using a Bi-LSTM-CRF. Lample et al. (2016) proposed two models which incorporated Bi-LSTM-composed character embeddings alongside words: a Bi-LSTM-CRF, and a greedy stack LSTM which uses a simple shift-reduce grammar to compose words into labeled entities. Their Bi-LSTM-CRF obtained the state-of-the-art on four languages without word shape or lexicon features. Ma and Hovy (2016) use CNNs rather than LSTMs to compose characters in a Bi-LSTM-CRF, achieving state-of-the-art performance on part-of-speech tagging and CoNLL NER without lexicons. Chiu and Nichols (2016) evaluate a similar network but propose a novel method for encoding lexicon matches, presenting results on CoNLL and OntoNotes NER. Yang et al. (2016) use GRU-CRFs with GRU-composed character embeddings of words to train a single network on many tasks and languages.

In general, distributed representations for text can provide useful generalization capabilities for NER systems, since they can leverage unsupervised pre-training of distributed word representations (Turian et al., 2010; Collobert et al., 2011; Passos et al., 2014). Though our models would also likely benefit from additional features such as character representations and and lexicons, we focus on simpler models which use word-embeddings alone, leaving more elaborate input representations to future work.

In these NER approaches, CNNs were used for low-level mapping feature extraction that feeds into alternative architectures. Overall, end-to-end CNNs have mainly been used in NLP for sentence classification, where the output representation is lower resolution is lower than that of the input Kim (2014); Kalchbrenner et al. (2014); Zhang et al. (2015); Toutanova et al. (2015).

Our work draws on the use of dilated convolutions for image segmentation in the computer vision community (Yu and Koltun, 2016; Chen et al., 2015). Similar to our block, Yu and Koltun (2016) employ a *context-module* of stacked dilated convolutions of exponentially increasing dilation width. Dilated convolutions were recently applied to the task of speech generation (van den Oord et al., 2016), and concurrent with this work, Kalchbrenner et al. (2016) posted a pre-print describing a network for machine translation that uses dilated convolutions in the encoder and decoder components. We are the first to use dilated convolutions for sequence labeling.

The broad receptive field of the ID-CNN helps aggregate document-level context. Ratinov and Roth (2009) incorporate document context in their greedy model by adding features based on tagged entities within a large, fixed window of tokens. Prior work has also posed a structured model that couples predictions across the whole document (Bunescu and Mooney, 2004; Sutton and McCallum, 2004; Finkel et al., 2005).

## 6   Experimental Results

We describe experiments on two benchmark English named entity recognition datasets. On CoNLL-2003 English NER, our ID-CNN outperforms a Bi-LSTM as a feature extractor for a CRF, and with greedy decoding performs on-par with the Bi-LSTM-CRF while running at more

than 14 times the speed. We also observe a performance boost in almost all models when broadening the context to incorporate entire documents, achieving an average F1 of 90.65 on CoNLL-2003, out-performing the Bi-LSTM-CRF while decoding at nearly 8 times the speed.

## 6.1 Data and Evaluation

We evaluate using labeled data from the CoNLL-2003 shared task (Tjong Kim Sang and De Meulder, 2003) and OntoNotes 5.0 (Hovy et al., 2006; Pradhan et al., 2006). Following previous work, we use the same OntoNotes data split used for co-reference resolution in the CoNLL-2012 shared task (Pradhan et al., 2012). For both datasets, we convert the IOB boundary encoding to BILOU as previous work found this encoding to result in improved performance (Ratinov and Roth, 2009). As in previous work we evaluate the performance of our models using segment-level micro-averaged F1 score. Hyperparameters that resulted in the best performance on the validation set were selected via grid search. A more detailed description of the data, evaluation, optimization and data pre-processing can be found in the Appendix.

## 6.2 Baselines

We compare our **ID-CNN** against strong LSTM and CNN baselines: a **Bi-LSTM** with local decoding, and one with CRF decoding (**Bi-LSTM-CRF**). We also compare against a non-dilated CNN architecture with the same number of convolutional layers as our dilated network (**4-layer CNN**) and one with enough layers to incorporate a receptive field of the same size as that of the dilated network (**5-layer CNN**) to demonstrate that the dilated convolutions more effectively aggregate contextual information than simple convolutions (i.e. using fewer parameters). We also compare our document-level ID-CNNs to a baseline which does not share parameters between blocks (**noshare**) and one that computes loss only at the last block, rather than after every iterated block of dilated convolutions (**1-loss**).

We do not compare with deeper or more elaborate CNN architectures for a number of reasons: 1) Fast train and test performance are highly desirable for NLP practitioners, and deeper models require more computation time 2) more complicated models tend to over-fit on this relatively small dataset and 3) most accurate deep CNN architectures repeatedly up-sample and down-sample the

| Model | F1 |
|---|---|
| Ratinov and Roth (2009) | 86.82 |
| Collobert et al. (2011) | 86.96 |
| Lample et al. (2016) | 90.33 |
| Bi-LSTM | $89.34 \pm 0.28$ |
| 4-layer CNN | $89.97 \pm 0.20$ |
| 5-layer CNN | $90.23 \pm 0.16$ |
| ID-CNN | $90.32 \pm 0.26$ |
| Collobert et al. (2011) | 88.67 |
| Passos et al. (2014) | 90.05 |
| Lample et al. (2016) | 90.20 |
| Bi-LSTM-CRF (re-impl) | $90.43 \pm 0.12$ |
| ID-CNN-CRF | $\mathbf{90.54 \pm 0.18}$ |

Table 1: F1 score of models observing sentence-level context. No models use character embeddings or lexicons. Top models are greedy, bottom models use Viterbi inference

inputs. We do not compare to stacked LSTMs for similar reasons — a single LSTM is already slower than a 4-layer CNN. Since our task is sequence labeling, we desire a model that maintains the token-level resolution of the input, making dilated convolutions an elegant solution.

## 6.3 CoNLL-2003 English NER

### 6.3.1 Sentence-level prediction

Table 1 lists F1 scores of models predicting with sentence-level context on CoNLL-2003. The Viterbi-decoding Bi-LSTM-CRF and ID-CNN-CRF obtain the highest average scores, with the ID-CNN-CRF outperforming the Bi-LSTM-CRF by 0.11 points of F1 on average, and the Bi-LSTM-CRF out-performing the greedy ID-CNN by 0.11 as well. Our greedy ID-CNN outperforms all other greedy models, including the 4-layer CNN which uses the same number of parameters as the ID-CNN, and the 5-layer CNN which uses more parameters but covers the same size receptive field. All CNN models out-perform the Bi-LSTM when paired with greedy decoding, suggesting that CNNs are better feature extractors than Bi-LSTMs for independent logistic regression. When paired with Viterbi decoding, our ID-CNN out-performs the Bi-LSTM as a feature extractor, showing that the D-CNN is also a better feature extractor for Viterbi inference.

Our ID-CNN is not only a better feature extractor than the Bi-LSTM but it is also faster. Table 2 lists relative decoding times on the CoNLL de-

| Model | $b = 1$ | Fastest ($b$) |
|---|---|---|
| Bi-LSTM-CRF | $1\times$ | $1\times$ (1024) |
| ID-CNN-CRF | $4.55\times$ | $1.28\times$ (32) |
| Bi-LSTM | $1.01\times$ | $9.92\times$ (2048) |
| 5-layer CNN | $6.56\times$ | $12.38\times$ (2048) |
| ID-CNN | $6.00\times$ | $14.10\times$ (128) |

Table 2: Relative test-time speed of sentence models, using batch size $b = 1$, and the fastest batch size ($b$) for each model.[4]

velopment set, compared to the Bi-LSTM-CRF. We report decoding times for batch size 1, giving Viterbi-decoding algorithms the advantage since much of their computational overhead comes from single-thread decoding on the CPU, and with the fastest batch size for each model.[2]

The D-CNN model is about 6 times faster than the Bi-LSTM when decoding one sentence at a time, and with larger batch sizes is nearly 50% faster. With Viterbi decoding, the gap closes somewhat but the ID-CNN-CRF still comes out ahead, about 30% faster than the Bi-LSTM-CRF. The most vast speed improvements come when comparing the greedy ID-CNN to the Bi-LSTM-CRF – our ID-CNN is more than 14 times faster than the Bi-LSTM-CRF at test time, but only 0.11 F1 points less accurate. The 5-layer CNN, which observes the same size receptive field as the ID-CNN but with more parameters, performs at about the same speed as the ID-CNN while making less accurate predictions. With a better implementation of dilated convolutions than currently included in TensorFlow, we would expect the D-CNN to be notably faster than the 5-layer CNN.

We emphasize the importance of the dropout regularizer of Ma et al. (2017) in Table 3, where we observe increased F1 for every model trained with expectation-linear dropout regularization. Dropout is important for training neural network models that generalize well, especially on relatively small NLP datasets such as CoNLL-2003. We recommend this regularizer as a simple and helpful tool for practitioners training neural networks for NLP.

---

[2]At scale, speed should increase with batch size, as we could compose each batch of as many sentences of the same length as would fit in GPU memory, requiring no padding and giving CNNs and D-CNNs even more of a speed advantage.

[4]Our D-CNN could see up to $18\times$ speed-up with a less naive implementation than is included in TensorFlow as of this writing.

| Model | w/o DR | w/ DR |
|---|---|---|
| Bi-LSTM | $88.89 \pm 0.30$ | $\mathbf{89.34 \pm 0.28}$ |
| 4-layer CNN | $89.74 \pm 0.23$ | $\mathbf{89.97 \pm 0.20}$ |
| 5-layer CNN | $89.93 \pm 0.32$ | $\mathbf{90.23 \pm 0.16}$ |
| Bi-LSTM-CRF | $90.01 \pm 0.23$ | $\mathbf{90.43 \pm 0.12}$ |
| 4-layer D-CNN | $89.65 \pm 0.30$ | $\mathbf{90.32 \pm 0.26}$ |

Table 3: Comparison of models trained with and without expectation-linear dropout regularization (DR). DR improves all models.

| Model | F1 |
|---|---|
| 4-layer D-CNN (sent) | $90.32 \pm 0.26$ |
| Bi-LSTM-CRF (sent) | $90.43 \pm 0.12$ |
| 4-layer CNN $\times$ 3 | $90.32 \pm 0.32$ |
| 5-layer CNN $\times$ 3 | $90.45 \pm 0.21$ |
| Bi-LSTM | $89.09 \pm 0.19$ |
| Bi-LSTM-CRF | $90.60 \pm 0.19$ |
| ID-CNN | $\mathbf{90.65 \pm 0.15}$ |

Table 4: F1 score of models trained to predict document-at-a-time. Our greedy ID-CNN model performs as well as the Bi-LSTM-CRF.

### 6.3.2 Document-level prediction

In Table 4 we show that adding document-level context improves every model on CoNLL-2003. When incorporating document-level context, our greedy ID-CNN model out-performs the Bi-LSTM-CRF, attaining 90.65 average F1. We believe this model out-performs the Bi-LSTM-CRF due to the ID-CNN learning a feature function better suited for representing broad context, in contrast with the Bi-LSTM which, though better than a simple RNN at encoding long memories of sequences, may reach its limit when provided with sequences more than 1,000 tokens long such as entire documents.

We also note that our combination of training objective (Eqn. 10) and tied parameters (Eqn. 7) more effectively learns to aggregate this broad context than a vanilla cross-entropy loss or deep CNN back-propagated from the final neural network layer. Table 5 compares models trained to incorporate entire document context using the document baselines described in Section 6.2.

In Table 6 we show that, in addition to being more accurate, our ID-CNN model is also much faster than the Bi-LSTM-CRF when incorporating context from entire documents, decoding at almost 8 times the speed. On these long sequences, it also

| Model | F1-avg | F1-max |
|---|---|---|
| ID-CNN noshare | $89.81 \pm 0.19$ | 89.75 |
| ID-CNN 1-loss | $90.06 \pm 0.19$ | 90.15 |
| ID-CNN | $\mathbf{90.65 \pm 0.15}$ | **90.66** |

Table 5: Comparing ID-CNNs with 1) back-propagating loss only from the final layer (**1-loss**) and 2) untied parameters across blocks (**noshare**)

| Model | $b = 1$ | Fastest ($b$) |
|---|---|---|
| Bi-LSTM-CRF | $1\times$ | $1\times$ (1024) |
| Bi-LSTM | $1.05\times$ | $4.60\times$ (1024) |
| ID-CNN | $40.83\times$ | $7.96\times$ (32) |

Table 6: Relative test-time speed of document models, using batch size $b = 1$, and the fastest batch size ($b$) for each model.

tags at more than 4.5 times the speed of the greedy Bi-LSTM, demonstrative of the benefit of our ID-CNNs context-aggregating computation that does not depend on the length of the sequence.

### 6.4 OntoNotes 5.0 English NER

We observe similar patterns on OntoNotes as we do on CoNLL. Table 7 lists overall F1 scores of our models compared to those in the existing literature. The greedy Bi-LSTM out-performs the lexicalized greedy model of Ratinov and Roth (2009), and our ID-CNN out-performs the Bi-LSTM as well as the more complex model of Durrett and Klein (2014) which leverages the parallel co-reference annotation available in the OntoNotes corpus to predict named entities jointly with entity linking and co-reference. Our greedy model is out-performed by the Bi-LSTM-CRF reported in Chiu and Nichols (2016) as well as our own re-implementation, which appears to be the new state-of-the-art on this dataset.

| Model | F1 |
|---|---|
| Ratinov and Roth (2009)[5] | 83.45 |
| Durrett and Klein (2014) | 84.04 |
| Chiu and Nichols (2016) | $86.19 \pm 0.25$ |
| Bi-LSTM | $83.76 \pm 0.10$ |
| ID-CNN | $84.23 \pm 0.30$ |
| Bi-LSTM-CRF | $86.99 \pm 0.22$ |
| Bi-LSTM-CRF (doc) | $86.81 \pm 0.18$ |
| ID-CNN (doc) | $85.76 \pm 0.13$ |

Table 7: F1 score of sentence and document models on OntoNotes.

The gap between our greedy model and those using Viterbi decoding is wider than on CoNLL. We believe this is due to the more diverse set of entities in OntoNotes, which also tend to be much longer – the average length of a multi-token named entity segment in CoNLL is about one token shorter than in OntoNotes. These long entities benefit more from explicit structured constraints enforced in Viterbi decoding. Still, our ID-CNN outperforms all other greedy methods, achieving our goal of learning a better feature extractor for structured prediction.

Incorporating greater context significantly boosts the score of our greedy model on OntoNotes, whereas the Bi-LSTM-CRF performs more poorly. In Table 7, we also list the F1 of our ID-CNN model and the Bi-LSTM-CRF model trained on entire document context. For the first time, we see the score decrease when more context is added to the Bi-LSTM-CRF model, though the ID-CNN, whose sentence model has a much lower score than that of the Bi-LSTM-CRF, sees an increase of more than 1.5 F1 points. We believe the decrease in the Bi-LSTM-CRF model occurs because of the nature of the OntoNotes dataset compared to CoNLL-2003: CoNLL-2003 contains a particularly high proportion of ambiguous entities,[6] perhaps leading to more benefit from document context that helps with disambiguation. In this scenario, adding the wider context may just add noise to the high-scoring Bi-LSTM-CRF model, whereas the less accurate dilated model can still benefit from the refined predictions of the iterated dilated convolutions.

## 7 Conclusion

We present iterated dilated convolutional neural networks, fast feature extractors that efficiently aggregate broad context without losing resolution. These provide impressive speed improvements for sequence labeling, particularly when processing entire documents at a time. In the future we hope to extend this work to NLP tasks with richer structured output, such as parsing.

---

[5]Results as reported in Durrett and Klein (2014) as this data split did not exist at the time of publication.

[6]According to the ACL Wiki page on CoNLL-2003: "The corpus contains a very high ratio of metonymic references (city names standing for sport teams)"

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
