# Peer review of "Fast and Accurate Sequence Labeling with Iterated Dilated Convolutions"

_ACL 2017 — decision unknown_

[Official Review · Reviewer 1 · rating 4 · confidence 5]
soundness 3 · originality 4 · clarity 4 · impact 3 · substance 3 · appropriateness 5 · meaningful comparison 5 · presentation format Oral Presentation

This work proposes to apply dilated convolutions for sequence tagging
(specifically, named entity recognition). It also introduces some novel ideas
(sharing the dilated convolution block, predicting the tags at each convolution
level), which I think will prove useful to the community. The paper performs
extensive ablation experiments to show the effectiveness of their approach.
I found the writing to be very clear, and the experiments were exceptionally
thorough.

Strengths:  
- Extensive experiments against various architectures (LSTM, LSTM + CRF)       
- Novel architectural/training ideas (sharing blocks)  

Weaknesses:  
- Only applied to English NER--this is a big concern since the title of the
paper seems to reference sequence-tagging directly.  
- Section 4.1 could be clearer. For example, I presume there is padding to make
sure the output resolution after each block is the same as the input
resolution.  Might be good to mention this.  
- I think an ablation study of number of layers vs perf might be interesting.

RESPONSE TO AUTHOR REBUTTAL:

Thank you very much for a thoughtful response. Given that the authors have
agreed to make the content be more specific to NER as opposed to
sequence-tagging, I have revised my score upward.

[Official Review · Reviewer 2 · rating 3 · confidence 4]
soundness 3 · originality 4 · clarity 2 · impact 3 · substance 3 · appropriateness 4 · meaningful comparison 5 · presentation format Poster

- Strengths:

The main strength promised by the paper is the speed advantage at the same
accuracy level.

- Weaknesses:

Presentation of the approach leaves a lot to be desired. Sections 3 and 4 need
to be much clearer, from concept definition to explaining the architecture and
parameterization. In particular Section 4.1 and the parameter tieing used need
to be crystal clear, since that is one of the main contributions of the paper.

More experiments supporting the vast speed improvements promised need to be
presented. The results in Table 2 are good but not great. A speed-up of 4-6X is
nothing all that transformative.

- General Discussion:

What exactly is "Viterbi prediction"? The term/concept is far from established;
the reader could guess but there must be a better way to phrase it.

Reference Weiss et al., 2015 has a typo.